# Role of Mediterranean Diet and Ultra-Processed Foods on Sperm Parameters: Data from a Cross-Sectional Study

**DOI:** 10.3390/nu17132066

**Published:** 2025-06-21

**Authors:** Gabriel Cosmin Petre, Francesco Francini-Pesenti, Luca De Toni, Andrea Di Nisio, Asia Mingardi, Ilaria Cosci, Nicola Passerin, Alberto Ferlin, Andrea Garolla

**Affiliations:** 1Unit of Andrology and Reproductive Medicine, Centre for Male Gamete Cryopreservation, Department of Medicine, University of Padova, 35122 Padova, Italy; gabriel.petre@rocketmail.com (G.C.P.); luca.detoni@unipd.it (L.D.T.); asia.mingardi@studenti.unipd.it (A.M.); ilaria.cosci@unipd.it (I.C.); nicola.passerin@aopd.veneto.it (N.P.); alberto.ferlin@unipd.it (A.F.); 2Clinical Nutrition Unit, Department of Medicine, University of Padova, 35122 Padova, Italy; francescofrancini@yahoo.it; 3Department of Health, Nutrition and Sport, Pegaso Telematic University, 80143 Naples, Italy; andrea.dinisio@unipegaso.it

**Keywords:** Mediterranean diet, fertility, nutrition, sperm parameters, diet, infertility, nutrients, ultra-processed food

## Abstract

**Background/Objectives:** Male infertility is multifactorial, involving genetic, environmental, lifestyle, and medical factors. Recent research has underscored the influence of lifestyle choices, such as dietary habits, smoking, alcohol abuse, and metabolic disturbances, on sperm quality. In this context, nutrition plays a pivotal role: adherence to a healthy diet like the Mediterranean Diet (MD), which emphasizes seasonal, fresh, and whole foods, has been linked to improved sperm performance. Conversely, a high intake of ultra-processed foods (UPFs), characterized by additives, high levels of sugars, fats, and salt, and a nutrient-poor profile, may impair sperm quality. **Methods:** Based on data supporting the reproductive health benefits of the MD, this observational cross-sectional study aimed at evaluating the possible relationship between MD adherence, assessed using the 14-point *a priori* Mediterranean Diet Adherence Screener (MEDAS) and intake of ultra-processed foods (UPFs), based on the NOVA classification, and sperm quality in 358 individuals (mean age 34.6 ± 9.3 years) who spontaneously referred to our center of reproductive medicine. Semen analyses were performed according to the WHO 2021 criteria. Hormonal profiles (FSH, LH, testosterone, SHBG, bioavailable testosterone, and calculated free testosterone) were also determined. **Results**: MD adherence score was significantly and positively correlated with semen parameters, whilst negatively correlated with FSH and LH levels. In contrast, UPF intake was correlated with poor semen parameters, whilst no association was observed with hormonal levels. Multivariate analyses confirmed these associations and showed the independency from age and BMI. Notably, among men with FSH levels < 8 IU/mL, higher quartiles of UPF intake had lower markers of sperm quality, particularly for viability and typical morphology. Differently, high MD adherence scores were associated with high quality sperm parameters even when FSH levels were >8 IU/mL. **Conclusions:** This study provides evidence that the adherence to MD, and conversely reduced intake of ultra-processed foods, is associates with a better semen profile. These findings suggest the possible role of dietary interventions as a modifiable factor in the management of male infertility.

## 1. Introduction

Infertility, often referred to as subfertility, is a medical condition characterized by the inability of a couple to achieve pregnancy after 12 months of regular and unprotected sexual intercourse. It affects approximately 15 to 25% of couples worldwide, resulting in about 186 million people worldwide [1,2]. In particular, about 40–50% of cases recognize a male factor and may result from issues with sperm production, sperm function, or sperm delivery [3,4]. Male infertility is often multifactorial, involving a combination of genetic, environmental, medical conditions and lifestyle habits, such as cigarette smoking, physical inactivity, alcohol abuse and dietary habits [5,6]. These lifestyle-related factors, can negatively affect the quantity, motility, morphology and genetic pattern of sperm, as well as overall reproductive health [7,8]. In addition, many metabolic derangements, such as weight excess, insulin resistance/impaired glucose metabolism, and determinants of metabolic syndrome, may have a negative impact on male fertility [9,10].

In this frame, the key role of nutrition in male fertility potential should be emphasized since as the adherence to an healthy dietary pattern, such as the Mediterranean Diet (MD) model, has been associated with better semen parameters and sperm function [11,12,13]. On the other hand, little is known about the possible association between ultra-processed foods (UPFs) and male fertility [14,15]. The optimal intake of some vitamins, minerals, and antioxidants, especially from food sources, is recognized to have an essential role in spermatogenesis [16,17,18]. The MD pattern, based on ancient traditional dietary patterns of countries bordering the Mediterranean Sea, provides an appropriate supply of the aforementioned oligo-nutrients [19,20]. Differently, UPFs are industrially manufactured food products that contain ingredients not typically found in natural foods, such as additives, artificial flavorings, emulsifiers, preservatives, and high levels of refined sugars, fats, and salt. These ingredients are the result of extensive plant processing that reduces the original content of essential nutrients whilst adding artificial components in order to enhance shelf life, texture, and flavor [21,22]. Thus, these products are calorie-dense but nutrient-poor foods, lacking essential vitamins, minerals, and fiber. Increasing literature underlines that the consumption of UPFs is associated with a higher risk of non-communicable diseases and even cancer [21,23].

Based on the large amount of data confirming the positive effects of MD on various disease risk, that greater adherence to this dietary pattern is suggested to have benefits in terms of semen parameters and reproductive capacity, and vice versa, a high intake of UPFs might negatively affect semen parameters. In the present study, we evaluated the possible correlation between the adherence to the MD model, as opposed to UPF intake, and the quality of sperm parameters in subjects undergoing semen analysis.

## 2. Materials and Methods

### 2.1. Study Population and Nutritional Habits

The aim of this observational cross-sectional study was to evaluate the relationships between sperm parameters and nutritional habits in adult male subjects. A total of 358 participants in the age range 18–60 years who attended the Unit of Andrology and Reproductive Medicine—University Hospital of Padova (Italy) for semen analysis assessment, were prospectively recruited during the time-frame September 2022–April 2024, at their first outpatient evaluation, upon signing the informed consent (Territorial Ethics Committee Central-Eastern Veneto Area, study no. AOP3205). Exclusion criteria included the following: current treatment with medications that could influence sexual and/or reproductive function, current adherence to a nutritional intervention or completed within the previous 6 months, including dietary supplements, liver failure, chronic inflammatory bowel diseases, type 1 (T1DM) and/or type 2 diabetes mellitus (T2DM), endocrinopathies, chronic renal failure with an estimated glomerular filtration rate < 60 mL/min, recent history of major cardiovascular events, eating disorders or other established psychiatric disorders, inflammation and/or infections in the genitourinary tract and/or undergoing treatment for such conditions, history of febrile episodes with a temperature ≥ 38 °C for at least two days in the previous 3 months, previous neoplasms, confirmed genetic causes of infertility and failure to comply with the required period of sexual abstinence for seminal analysis and smokers. Since physical activity may affect male fertility, subjects self-reporting physical training sessions for more than 3 h per week were excluded [24].

Subjects underwent an outpatient evaluation that included medical history recording, anthropometric evaluation, smoking habit, and dietary assessment. Body mass index (BMI) was calculated as the ratio of body weight (in kg) to the square of height (in meters). Semen collection was obtained by masturbation into sterile containers after 3–7 days of sexual abstinence, and analysis was performed according to the WHO 2021 criteria [25]. After collection, semen samples were maintained at 37 °C for at least 30 min before analysis and then evaluated for semen volume, pH, sperm concentration, total sperm count, cell viability, and typical cell morphology. Sperm concentration was assessed using the Makler counting chamber at least two 50 µL aliquots in order to ensure accuracy and minimize random errors. All analyses adhered to standard procedures of our laboratory and were performed by the same experienced operator. Circulating plasma levels of total testosterone, follicle-stimulating hormone (FSH), luteinizing hormone (LH), and sex hormone-binding globulin (SHBG) were tested as previously described [26].

MD adherence was evaluated through the validated questionnaire Mediterranean Diet Adherence Screener (MEDAS) [27]. Briefly, this instrument involves 14 items focused on the main food categories related to MD. MEDAS questionnaire was validated through the 136-item food frequency questionnaire (FFQ) and was recognized as a valid tool for the rapid estimation of MD adherence [27]. MEDAS value evaluation has also been further categorized into three levels of adherence to MD: ≤5, low adherence; 6–9 = medium adherence; and ≥10, high adherence [27]. To assess the intake of UPFs, we used a 24-h Food Recall Questionnaire, a method for evaluating the daily intake of ultra-processed foods based on the NOVA classification criteria [28]. The two dietary assessments were performed by the same dietitian. Semen analysis was conducted blindly to the results of the dietary assessment.

### 2.2. Statistics

All statistical analyses were conducted using SPSS software version 25.0 (SPSS Inc., Chicago, IL, USA). Data were collected in a pseudo-anonymized format on digital support, and the analyzed variables were computed as categorical or continuous variables. The Shapiro–Wilk test was used to address the normal distribution of continuous parameters. Differences between continuous variables were analyzed using ANOVA. Differences between discrete variables were analyzed using the Chi-square test or Fisher’s exact test (if the expected count was <5).

To describe correlations between variables, Pearson’s correlation coefficient, *r*, was used for normally distributed variables. Based on the correlation analyses, multivariate analyses were performed to compare seminal parameters most strongly associated with MEDAS scores and ultra-processed food intake, adopting Bonferroni’s post hoc test for pairwise comparisons between subgroups. The association between a set of continuous and ordinal variables with sperm count ≥ 39 million/ejaculate dichotomous outcome was evaluated by logistic regression analysis, and the respective odds ratios (ORs) with 95% confidence intervals (CIs) was calculated. Statistical significance was defined for *p* < 0.05.

## 3. Results

Demographic, anthropometric, and clinical characteristics of the 358 male subjects finally included in the study are reported in Table 1. Mean age and BMI were 34.6 ± 9.3 years and 24.4 ± 4.2 Kg/m^2^, respectively, showing a general trend towards the young age and normal weight in the study group. The mean MEDAS value of the whole sample was suggestive of an average adherence to MD. Mean values of sperm parameters were within the normal range, although with broad SD values.

A preliminary bivariate correlation analysis was performed between nutritional pattern scores, MEDAS, and the percentage of total calories derived from ultra-processed foods (%Kal-UP), FSH, and demographic, seminal, and hormonal parameters, whose results are reported in Table 2. An extremely significant and positive correlation was observed between %Kal-UP and the percentage of calories from ultra-processed sources within each macro-nutrients category: carbohydrates, proteins, and lipids. For this reason, %Kal-UP was used as a representative parameter for ultra-processed food intake in subsequent analyses. The MEDAS values showed a significant negative correlation with %Kal-UP and positive correlation with all seminal parameters, with the exception of semen pH. A significant and negative correlation was also observed with serum levels of FSH and LH. Conversely, %Kal-UP showed a significant and negative correlation with all semen parameters, but no correlation with hormonal parameters was observed. Neither MEDAS nor %Kal-UP were correlated with age or BMI, and all the aforementioned correlations between MEDAS or %Kal-UP and semen parameters were maintained after correction for the patient’s age (all *p* values < 0.001). A further bivariate correlation analysis with MEDAS was performed, by controlling for, respectively, FSH, %Kal-UP, and BMI as major confounders. Whilst significant correlations were maintained after controlling for BMI, the significant correlations between MEDAS and clinical parameters were lost by controlling for FSH or %Kal-UP, suggesting a mutual interaction among the three parameters.

Subjects were then stratified into recognized categories of MD adherence, as previously described by García-Conesa et al., according to MEDAS values and respectively: low adherence for a score ≤ 5 (group 1), moderate adherence for a score between 6 and 9 (group 2), and high adherence for a score ≥ 10 (group 3) [27]. Multivariate analysis showed that, with the exception for sperm volume, highly significant differences in sperm parameters were observed across the three categories of adherence to MD, with increasing mean values of sperm concentration, total sperm count, progressive motility, sperm viability, and the percentage of sperm with typical morphology along with the adherence to MD (all *p*-values < 0.001, Figure 1A). On the other hand, a significant reduction of LH levels was observed between MEDAS categories 1 and 2, whilst reduced levels of FSH were found in categories 2 and 3 compared to category 1 (all *p*-values < 0.05, Figure 1B). No significant differences were observed in SHBG and testosterone levels, whether total, bioavailable, or free.

A similar approach was adopted for the intake of ultra-processed foods. However, since no discrete categories of ultra-processed food intake are available as for the MEDAS score, the unbiased stratification by increasing quartiles (Q) of %Kal-UP was performed (respectively: Q1 0.5–10.8%, Q2 10.8–24.1%, Q3 24.1–42.6%, Q4 42.6–96.6%. Figure 2A,B). Except for semen volume, a significant reduction of mean values of sperm concentration, total sperm count, progressive motility, and percentage of cells with typical morphology was observed by moving from Q1 to Q4 of %Kal-UP (all *p*-values < 0.01, Figure 2A). In addition, the mean value of the percentage of viable cells pertaining to Q1 of %Kal-UP was significantly lower than the mean values in the other quartiles. No significant variations were observed for hormonal parameters (Figure 2B). Further analyses showed that patients pertaining to category 1 of the MEDAS value and Q4 in %Kal-UP had significantly lower mean values of semen parameters compared to patients pertaining to category 3 of MEDAS value and Q1 in %Kal-UP (Figure 3). No significant difference was observed for hormonal levels.

### Evaluation of the Possible Effect of the Interaction Between Dietary Regimen and Hormonal Parameters on Semen Parameters by Multivariate Analysis

To clarify the possible role of the interaction between hormonal parameters and dietary regimen, in terms of MD adherence or UPFs intake, in affecting semen parameters, a multivariate analysis was performed. Patients were stratified according to serum FSH levels < 8 IU/mL or ≥8 IU/mL, representing a clinically relevant threshold level for normal and high FSH level, and further divided into quartiles of %Kal-UP (Figure 4). Among subjects with FSH < 8 IU/mL, the mean values of sperm concentration, total count, percentage of progressive motile cells, and percentage of immotile cells showed a progressive and significant reduction with increasing %Kal-UP quartiles, compared to quartile 1. Differently, subjects with FSH ≥ 8 IU/mL showed no effect on the aforementioned semen parameters according to %Kal-UP quartiles pertinence. However, for sperm viability, a significant reduction was found between Q1 and Q4 of %Kal-UP in subjects with FSH < 8 IU/mL and between Q2 and Q4 in subjects with FSH ≥ 8 IU/mL. Similarly, for typical morphology, a progressive reduction was noted in both FSH groups. A similar approach was used to evaluate the possible effect on semen parameters associated with the interaction between serum FSH levels and MEDAS values, reported as the respective ordinal variables, FSH levels < 8 IU/mL and MEDAS score categories, respectively (Figure 5). A significant increase in sperm concentration and total sperm count was observed with higher MD adherence, but only in subjects with FSH < 8 IU/mL. However, progressive motility, percentage of non-motile cells, and typical morphology showed a significant increase with increasing MD adherence categories in both FSH < 8 IU/mL and FSH ≥ 8 IU/mL groups.

Finally, a logistic regression analysis was performed to estimate the relative impact of MD adherence categories and %Kal-UP quartiles in associating with a sperm count lower than 39 × 10^6^ cells/ejaculate (Table 3). Categories of MD adherence 2 and 3 were associated with a significantly reduced risk of low sperm count (respectively: OR = 0.312, *p* = 0.001 and OR = 0.250, *p* = 0.027), whereas quartiles 3 and 4 of %Kal-UP were associated with significant increased risk of having a total sperm count lower than 39 × 10^6^ cells/ejaculate (respectively: OR = 4.469, *p* = 0.020 and OR = 3.490, *p* = 0.042). However, when FSH levels ≥ 8 U/mL were included in the model, this resulted as the only and highly significant factor associated with a reduced total sperm count (OR = 9.761, *p* = 0.001).

## 4. Discussion

In this study, we provide evidence that adherence to the MD and the consumption of ultra-processed foods significantly correlate with sperm parameters, independent from BMI. Though remarking the primary role of FSH in determining semen parameters, these findings suggest the possible role of nutritional regimens as a potential modifiable risk factor for semen impairment in patients without additional infertility determinants.

It is well known that fertility, particularly male fertility, is influenced by several environmental and lifestyle factors, including smoking, environmental pollution, alcohol consumption, low physical activity, and, most importantly, dietary habits [29,30,31,32].

The Mediterranean dietary pattern can be mechanistically associated with male fertility status by relying to fresh seasonal products and thus providing adequate levels of monounsaturated fatty acids and polyunsaturated fatty acids (PUFA), such as omega-3 and omega-6, together with an optimal supply of antioxidants, minerals, and vitamins [33,34]. On these bases, MD is thought to support optimal sperm maturation, as dietary fatty acids are key constituents of sperm cell membranes, affecting fluidity, capacitation, and acrosomal function [35,36]. Accordingly, a dietary pattern low in saturated fats and cholesterol, with a balanced ratio of polyunsaturated fatty acids (PUFA), appears to enhance sperm quality by maintaining cell membrane fluidity, reducing oxidative stress and inflammation, and improving mitochondrial function [37,38]. In this frame, it is crucial to note that inflammation and oxidative stress can influence male reproductive potential by inducing anatomical and functional changes in accessory glands. Dietary habits are thus believed to have a favorable impact on seminal vesicles and prostate health, reducing inflammatory processes and consequently improving sperm parameters [39,40].

Our results highlight an interplay association between FSH plasma levels, adherence to MD, UPFs intake, and semen parameters. In the presence of high FSH levels, suggestive of primary testicular impairment, the nutritional pattern is suggested to have mild or even no association with semen parameters. In particular, a high adherence to MD showed minor association with increased sperm progressive motility and typical morphology. In this condition, also UPFs intake shows less association with sperm parameters, except for cell viability. In contrast, when testicular function is preserved, as suggested by FSH levels < 8 U/mL, a greater adherence to the MD is significantly and positively associated with better seminal profile, whilst UPFs intake shows almost diametrically opposite association. Therefore, in the presence of a normal gonadal function three possible mechanistic scenarios can be hypothesized (Figure 6): (i) MD adherence associates with a better profile of the hypothalamic-pituitary-gonadal axis (HPTA) and testicular function; (ii) MD adherence likely associates with a better testicular sensitivity to FSH, resulting in a correct spermatogenic function and semen profile; (iii) MD might influence sperm production and spermiogenesis by the association with a reduced negative feedback at the central level. However, the cross-sectional nature of the present study does not allow us to address whether this evidence depends on testicular integrity, central function, or specific nutritional supply to germ cells. The latter hypothesis suggests that fertility markers might benefit from adequate nutritional supply of elements essential to spermatogenesis [41,42,43].

Our data shows that the combination of low MD adherence and UPFs intake is negatively associated with FSH and LH levels. However, no correlation is present with calculated free and bioavailable testosterone levels. Usually, an incorrect diet and high BMI are related to an increased rate of hypogonadism. In our sample, this observation is lacking probably because the mean BMI value of patients within the range of normal weight, contributes to maintaining testosterone levels within normal reference limits. In addition, hormonal parameters were available in a subset of patients, providing some numerical bias.

In a study by Karayiannis et al., evaluating 225 male subjects from couples attending a fertility clinic, authors showed that men with the lowest adherence to the MD had poorer sperm count, total motility, and sperm concentration, whereas higher adherence was significantly associated with better sperm quality [44]. Ricci et al. assessed Mediterranean diet adherence in 309 male partners of subfertile couples undergoing assisted reproductive techniques. Authors found a positive association between the Mediterranean diet and both sperm concentration and count, whilst no association was observed for semen volume [33]. However, these studies did not exclude other causes of seminal alteration, introducing a potential bias in the evaluation of dietary influence. Moreover, a systematic review by Salas-Huetos et al. found that diets rich in food groups such as fish, legumes, cereals, vegetables, and fruits and low in saturated fats and ultra-processed foods were positively associated with a better semen profile [7].

Compared to previous findings, our data are strengthened by the considerable sample size, homogeneous dietary assessment, and exclusion of known infertility causes, supporting an independent role of diet in sperm quality. In addition, we also considered physical activity as a confounding factor. Physical activity has a significant impact on male fertility, influencing key parameters such as sperm quality, hormonal profile, and oxidative stress. However, the effect depends on the intensity, duration, and type of exercise practiced. In fact, a recent meta-analysis suggests that moderate levels of physical activity are associated with better sperm quality, while intense physical activity may have negative effects on fertility by increasing oxidative stress [45]. We thus decided to exclude from our sample those who practiced more than 3 h of physical activity per week, to avoid that results were affected by this factor.

We acknowledge that the cross-sectional and observational nature of this study represents a limitation. In addition, although widely adopted in most studies, a single food intake assessment by the 24 h recall method has intrinsic limitation, being associated with large errors in estimating intake when compared with direct observation of the foods consumed (coefficient of differences ranged from 4 to 400%) [46,47]. Nonetheless, UPFs intake evaluation is largely estimated by a single assessment with a 24-h Food Recall Questionnaire [48]. Accordingly, also considering the documented criticisms in maintaining adequate patient participation in the follow-up, the study protocol involved a single dietary assessment at baseline [49]. Another limitation to our findings is represented by low number of patients for whom hormonal data were available. Accordingly, in order to adequately address the interaction between FSH and dietary regimens, the logistic regression analysis has been specifically split into two models, excluding or including FSH levels ≥ 8 IU/mL, respectively. The assessment of physical activity might have also clarified the role of a lifestyle factor potentially involved in fertility. Moreover, seminal parameters alone cannot be directly translated into fertility outcomes. Therefore, further prospective observational studies and well-designed clinical trials are recommended to expand on and better confirm our findings, addressing the possible role of nutrition as feasible approach to support male fertility. 

In conclusion, this study confirms that a greater adherence to MD and a lower consumption of UPFs are associated with a better semen profile, particularly in subjects with a preserved testicular function. This effect seems to be independent from age and even BMI. Additionally, a high MEDAS value was associated with lower FSH and LH levels. On the other hand, UPFs intake showed lower level of association.

## Figures and Tables

**Figure 1 nutrients-17-02066-f001:**
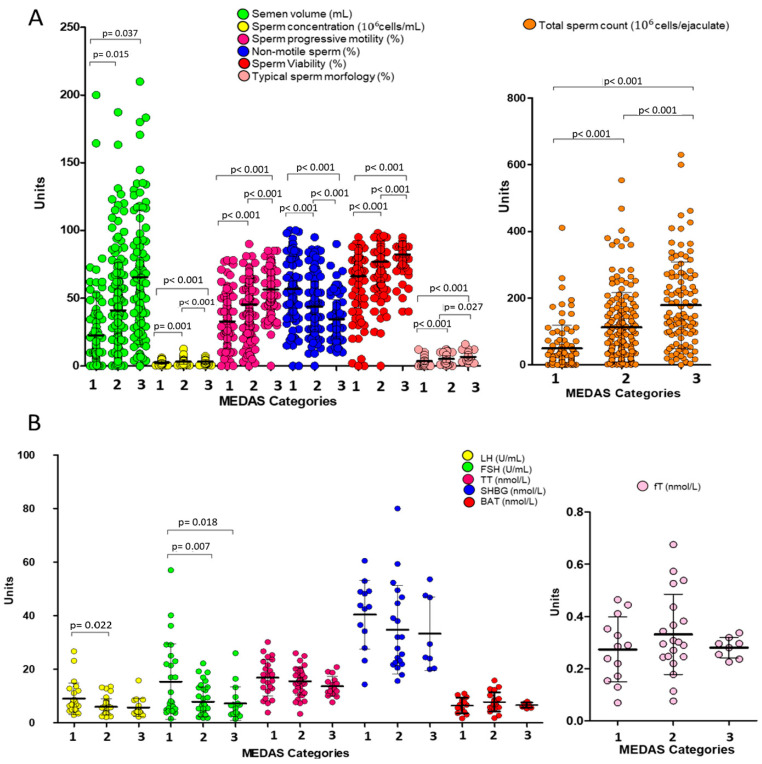
Effect of adherence to the Mediterranean diet, categorized by MEDAS score into low (1), moderate (2), and high adherence (3), on seminal (**A**) and hormonal parameters (**B**). FSH: follicle-stimulating hormone; LH: luteinizing hormone; TT: total testosterone; SHBG: sex hormone-binding globulin; BAT: bioavailable testosterone; fT: calculated free testosterone. The *p*-values associated with the comparison between pairs of groups are specified in the image.

**Figure 2 nutrients-17-02066-f002:**
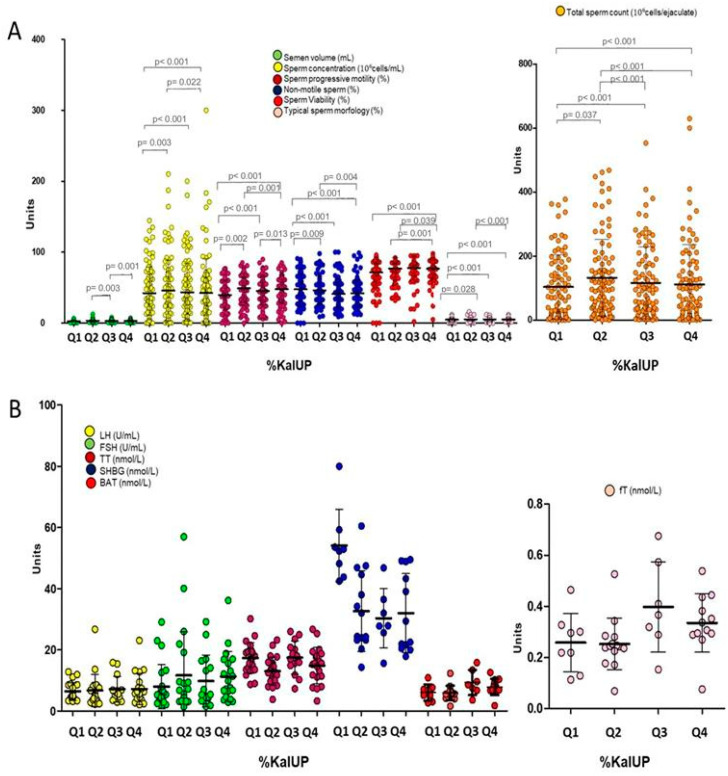
Effect of ultra-processed food consumption, assessed using the “24-h Food Recall Questionnaire” and categorized into low (Q1), medium-low (Q2), medium-high (Q3), and high consumption (Q4), on seminal (**A**) and hormonal parameters (**B**). FSH: follicle-stimulating hormone, LH: luteinizing hormone, TT: total testosterone, SHBG: sex hormone-binding globulin, BAT: bioavailable testosterone, fT: calculated free testosterone, %Kal-UP: percentage of calories derived from ultra-processed foods. The *p*-values associated with the comparison between pairs of groups are specified in the image.

**Figure 3 nutrients-17-02066-f003:**
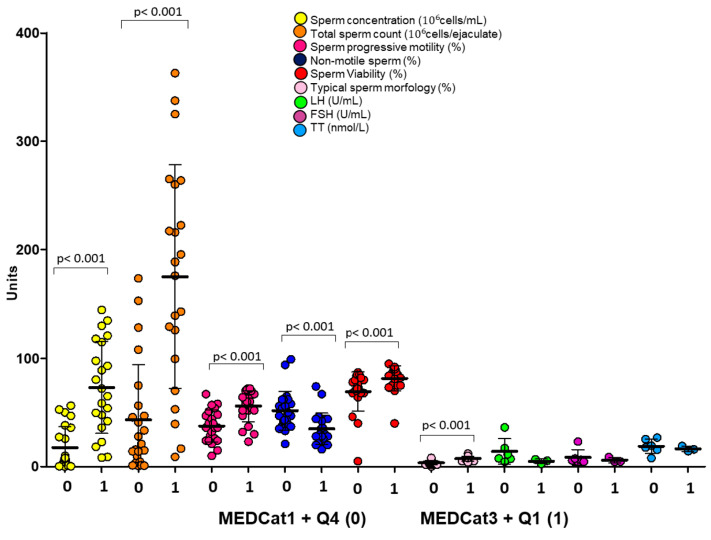
Comparison of seminal and hormonal parameters between subjects with low adherence to the Mediterranean diet (MEDCat1) and high consumption of ultra-processed foods (Q4) and subjects with high adherence to the Mediterranean diet (MEDCat3) and low consumption of ultra-processed foods (Q1) (respectively 0 and 1 conditions in the plot). FSH: follicle-stimulating hormone, LH: luteinizing hormone, TT: total testosterone. The *p*-values associated with the comparison between pairs of groups are specified in the image.

**Figure 4 nutrients-17-02066-f004:**
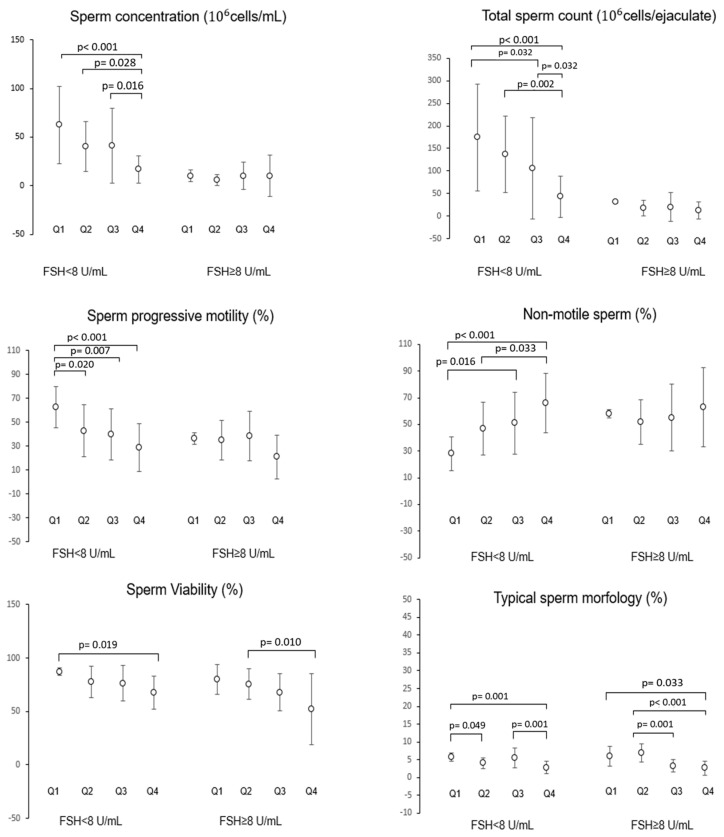
Evaluation of the interaction between gonadotropin levels, expressed as a dichotomous variable (FSH < 8 IU/mL and FSH ≥ 8 IU/mL), and the consumption of UPFs, categorized into low (Q1), medium-low (Q2), medium-high (Q3), and high consumption (Q4), on sperm parameters. FSH: Follicle-stimulating hormone. *p*-values associated with pairwise group comparisons are specified in the image.

**Figure 5 nutrients-17-02066-f005:**
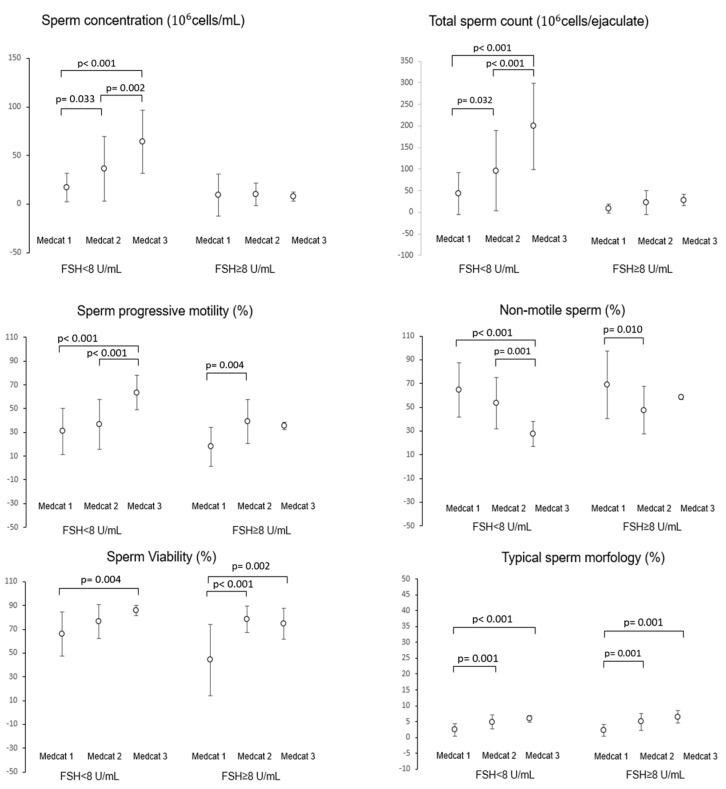
Evaluation of the interaction between gonadotropin levels, expressed as a dichotomous variable (FSH < 8 IU/mL and FSH ≥ 8 IU/mL), and MEDAS value categorized into low (Medcat1), medium (Medcat2), and high adherence (Medcat3), on seminal parameters. FSH: Follicle-stimulating hormone. The *p*-values associated with pairwise group comparisons are specified in the image.

**Figure 6 nutrients-17-02066-f006:**
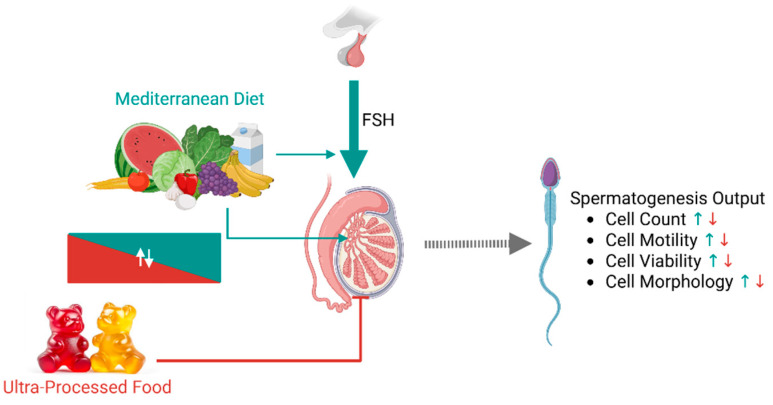
Representative scheme summarizing the possible mechanistic influences of dietary patterns on spermatogenesis. Adherence to Mediterranean diet (MD) likely has a direct positive effect on the different cell populations of the seminiferous tubules due to its composition rich in polyunsaturated fatty acids and oligo-elements. In parallel, MD might positively influence the hypothalamic-pituitary-gonadal axis by improving testis sensitivity to follicle-stimulating hormone (FSH) or by reducing the negative feedback at central level. Ultra-processed food intake negatively influences testis function by both being inversely related to MD adherence and because of the reduced content of protective factors. Green and red arrows indicate, respectively, positive and negative effects.

**Table 1 nutrients-17-02066-t001:** Sample characteristics. BMI: body mass index; UPFs: ultra-processed foods; FSH: follicle-stimulating hormone; LH: luteinizing hormone; SHBG: sex hormone-binding globulin; TT: total testosterone; BAT: bioavailable testosterone; fT: calculated free testosterone.

Parameter (Unit; Reference Values)	N	Mean ± SD	Min–Max
BMI (Kg/m^2^; 18.5–25 kg/m^2^)	358	24.4 ± 4.2	18.4–46.0
Age (years; N/A)	358	34.6 ± 9.3	18–67
MEDAS (N/A)	358	7.5 ± 2.8	2.5–13
Total Calories (Kcals; N/A)	358	1923.8 ± 344.9	1372–3965
Kcals from UPFs (%; N/A)	358	29.1 ± 21.9	0.64–96.6
Proteins Kcals from UPFs (%; N/A)	358	3.8 ± 2.7	0.1–9.7
Lipids Kcals from UPFs (%; N/A)	358	8.1 ± 17.4	0.16–28.9
Carbohydrates Kcals from UPFs (%; N/A)	358	17.4 ± 13.0	0.3–57.9
Sperm Volume (mL; >1.4 mL)	358	2.9 ± 1.5	0.2–12.5
Sperm pH (>7.2 and <8.0)	358	7.6 ± 0.2	6.4–8.0
Sperm Concentration (10^6^ cells/mL; >16 × 10^6^ cells/mL)	358	43.5 ± 42.3	n.d.–300.0
Sperm Count (10^6^ cells; >39 × 10^6^ cells)	358	116.6 ± 114.9	n.d.–629.8
Progressive Motility (%; >30%)	358	45.5 ± 20.7	0.0–90.0
Non-motile Sperm (%; <20%)	358	44.2 ± 20.5	9.0–100.0
Sperm Viability (%; >54%)	358	75.8 ± 17.5	2.0–98.0
Sperm Typical Morphology (%; >4%)	358	5.2 ± 2.7	0.0–16.0
FSH (U/mL; 1.5–9.4 U/mL)	80	10.2 ± 9.9	1.4–57
LH (U/mL; 1.5–9.4 U/mL)	76	6.8 ± 4.4	2.1–26.7
SHBG (nmol/L; 10–57 nmol/L)	41	36.2 ± 14.9	14.3–80.0
TT (nmoL/L; >10.4 nmol/L)	78	15.5 ± 5.5	3.3–30.2
BAT (nmoL/L; >2.4 nmol/L)	45	7.1 ± 3.1	1.6–19.42
fT (pmol/L; 229–1072 pmol/L)	41	302.7 ± 130.4	70.0–680.0

N/A: Not applicable; n.d.: Not determinable.

**Table 2 nutrients-17-02066-t002:** Correlations between clinical and demographic parameters and nutritional parameters in the 358 patients included in the study. BMI: body mass index, Tot Kcals: total calories intake, %Kal-UP: percentage of total calories intake derived from ultra-processed foods, %Prot Kal-UP: percentage of calories intake derived from ultra-processed proteins. %Lip Kal-UP: percentage of calories intake derived from ultra-processed lipids. %Cho Kal-UP: percentage of calories intake derived from ultra-processed carbohydrates. FSH: follicle-stimulating hormone. LH: luteinizing hormone. SHBG: sex hormone-binding globulin. TT: total testosterone. BAT: bioavailable testosterone. fT: calculated free testosterone. *r*: Pearson correlation coefficient. *p*: significance level. Statistically significant *p*-values are reported in bold.

	MEDAS	%Kal-UP	FSH	MEDAS	MEDAS	MEDAS
				Controlled for FSH	Controlled for %Kal-UP	Controlled for BMI
*Demographic/Anthropometric Data*						
BMI	*r* = −0.088 *p* = 0.096	*r* = 0.018 *p* = 0.736	*r* = 0.191 *p* = 0.090	*r* = 0.091 *p* = 0.863	*r* = −0.498 *p* = 0.315	*//*
Age	*r* = −0.009 *p* = 0.864	*r* = −0.075 *p* = 0.158	***r* = 0.262** ***p* = 0.019**	*r* = −0.138 *p* = 0.795	*r* = −0.454 *p* = 0.366	*r* = 0.337 *p* = 0.414
*Nutritional Data*						
MEDAS	//	***r* = −0.813** ***p* < 0.001**	***r* = −0.301** ***p* = 0.007**	**//**	*r* = −0.483 *p* = 0.331	*//*
Tot Kcals	***r* = −0.146** ***p* = 0.006**	*r* = 0.026 *p* = 0.622	*r* = 0.155 *p* = 0.169	*r* = −0.170 *p* = 0.747	//	*r* = −0.102 *p* = 0.530
%Kal-UP	***r* = −0.813** ***p* < 0.000**	**//**	*r* = 0.217 *p* = 0.053	*r* = −0.270 *p* = 0.605	//	***r* = −0.770** ***p* < 0.001**
%Prot Kal-UP	***r* = −0.803** ***p* < 0.001**	***r* = 0.960** ***p* < 0.001**	***r* = 0.253** ***p* = 0.023**	*r* = −0.218 *p* = 0.677	*r* = 0.207 *p* = 0.694	***r* = −0.810** ***p* < 0.001**
%Lip Kal-UP	***r* = −0.800** ***p* < 0.001**	***r* = 0.998** ***p* < 0.001**	*r* = 0.212 *p* = 0.059	*r* = −0.256 *p* = 0.625	*r* = 0.215 *p* = 0.682	***r* = −0.751** ***p* < 0.001**
%Cho Kal-UP	***r* = −0.814** ***p* < 0.001**	***r* = 0.999** ***p* < 0.001**	*r* = 0.211 *p* = 0.060	*r* = −0.291 *p* = 0.576	*r* = 0.242 *p* = 0.644	***r* = −0.764** ***p* < 0.001**
*Semen Parameters*						
Semen volume	***r* = 0.141** ***p* = 0.008**	***r* = −0.158** ***p* = 0.003**	*r* = 0.000 *p* = 1.000	*r* = 0.776 *p* = 0.070	*r* = 0.493 *p* = 0.320	***r* = 0.137** ***p* = 0.009**
Semen pH	***r* = −0.059** ***p* = 0.008**	*r* = 0.084 *p* = 0.113	*r* = −0.003 *p* = 0.982	*r* = 0.594 *p* = 0.214	*r* = −0.069 *p* = 0.897	*r* = −0.057 *p* = 0.286
Sperm concentration	***r* = 0.416** ***p* < 0.001**	***r* = −0.302** ***p* < 0.001**	***r* = −0.360** ***p* = 0.001**	*r* = −0.772 *p* = 0.072	*r* = 0.090 *p* = 0.865	***r* = 0.725** ***p* = 0.042**
Sperm count	***r* = 0.456** ***p* = 0.000**	***r* = −0.361** ***p* < 0.001**	***r* = −0.394** ***p* < 0.001**	*r* = −0.325 *p* = 0.530	*r* = 0.460 *p* = 0.359	***r* = 0.835** ***p* = 0.010**
Progressive motility	***r* = 0.431** ***p* < 0.001**	***r* = −0.365** ***p* < 0.001**	***r* = −0.308** ***p* = 0.005**	*r* = −0.085 *p* = 0.873	*r* = 0.432 *p* = 0.392	***r* = 0.721** ***p* = 0.043**
Non motile sperm	***r* = −0.390** ***p* < 0.001**	***r* = 0.322** ***p* < 0.001**	***r* = 0.276** ***p* = 0.013**	*r* = −0.024 *p* = 0.965	*r* = −0.506 *p* = 0.306	***r* = −0.768** ***p* = 0.021**
Sperm viability	***r* = 0.349** ***p* < 0.001**	***r* = −0.248** ***p* < 0.001**	***r* = −0.430** ***p* < 0.001**	*r* = 0.150 *p* = 0.777	*r* = 0.753 *p* = 0.084	***r* = 0.781** ***p* = 0.022**
Sperm morphology	***r* = 0.468** ***p* < 0.001**	***r* = −0.346** ***p* < 0.001**	***r* = −0.260** ***p* = 0.020**	*r* = −0.308 *p* = 0.552	*r* = 0.284 *p* = 0.585	***r* = 0.342** ***p* = 0.010**
*Hormonal Parameters*						
FSH	***r* = −0.301** ***p* = 0.007**	*r* = 0.217 *p* = 0.053	//	//	*r* = −0.811 *p* = 0.050	***r* = −0.745** ***p* = 0.034**
LH	*r* = −0.275 *p* = 0.16	*r* = 0.164 *p* = 0.158	***r* = 0.838** ***p* < 0.001**	*r* = −0.008 *p* = 0.988	*r* = −0.660 *p* = 0.153	***r* = −0.739** ***p* = 0.036**
SHBG	*r* = −0.116 *p* = 0.472	*r* = 0.166 *p* = 0.299	*r* = 0.241 *p* = 0.134	*r* = 0.163 *p* = 0.758	*r* = 0.380 *p* = 0.457	*r* = 0.635 *p* = 0.090
TT	*r* = −0.171 *p* = 0.135	*r* = 0.158 *p* = 0.166	*r* = −0.071 *p* = 0.543	*r* = −0.352 *p* = 0.493	*r* = 0.384 *p* = 0.452	***r* = 0.711** ***p* = 0.048**
BAT	*r* = 0.062 *p* = 0.702	*r* = −0.141 *p* = 0.379	***r* = −0.316** ***p* = 0.047**	*r* = −0.037 *p* = 0.822	*r* = −0.069 *p* = 0.670	*r* = 0.014 *p* = 0.933
fT	*r* = 0.062 *p* = 0.699	*r* = −0.141 *p* = 0.378	***r* = −0.316** ***p* = 0.047**	*r* = −0.036 *p* = 0.826	*r* = −0.069 *p* = 0.674	*r* = 0.013 *p* = 0.936

**Table 3 nutrients-17-02066-t003:** Results of logistic regression analysis conducted to estimate the most important factor in determining a total sperm count < 39 million/mL. Model 1 included categorized scores of the Mediterranean Diet Adherence Screener (MEDAS) questionnaire as low (MEDCat 1), medium (MEDCat 2), and high adherence (MEDCat 3), and the quartiles of percentage of caloric intake as ultra-processed foods (% Kal-UP), low (Q1), medium-low (Q2), medium-high (Q3), and high (Q4), respectively. In model 2, categorized follicle-stimulating hormone (FSH) as ≥8 U/mL was included. Odd ratios with 95% confidence interval (95% CI) are reported. Significant *p*-values are reported in bold.

Model	Variables	Odd Ratio	(95% C.I)	*p* Value
Model 1 Excluding FSH	MEDCat 1	Reference	//	0.003
	MEDCat 2	**0.312**	**(0.158–0.618)**	**0.001**
	MEDCat 3	**0.250**	**(0.073–0.856)**	**0.027**
	%Kal-UP Q1	Reference	//	0.045
	%Kal-UP Q2	1.833	(0.592–5.674)	0.293
	%Kal-UP Q3	**4.469**	**(1.269–15.740)**	**0.020**
	%Kal-UP Q4	**3.940**	**(1.050–14.786)**	**0.042**
	Constant	0.435		0.220
Model 2 Including FSH ≥ 8 U/mL	MEDCat 1	Reference	//	0.131
	MEDCat 2	0.512	(0.070–3.761)	0.510
	MEDCat 3	0.043	(0.002–1.100)	0.057
	%Kal-UP Q1	Reference	//	0.626
	% Kal-UP Q2	0.232	(0.017–3.136)	0.272
	% Kal-UP Q3	0.482	(0.029–8.053)	0.612
	% Kal-UP Q4	0.428	(0.017–11.107)	0.610
	FSH ≥ 8 U/mL	**9.761**	**(2.612–36.482)**	**0.001**
	Constant	3.494		0.457

## Data Availability

The data presented in this study are available on request from the corresponding author due to privacy and legal reasons.

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
