# Peer review of "Role of Mediterranean Diet and Ultra-Processed Foods on Sperm Parameters: Data from a Cross-Sectional Study"

_nutrients, 2025, doi:10.3390/nu17132066_

Round 1
Reviewer 1 Report (Previous Reviewer 2)
Comments and Suggestions for Authors
Comments to the Authors of manuscript number nutrients-3664237 entitled “Role of Mediterranean Diet and Ultra-processed Foods on Sperm Parameters: Data From a Cross-sectional Study”
- Although the authors consistently repeat that these are only correlations, in several places in the discussion and abstract they formulate statements suggesting the effect of diet on semen parameters. However, in cross-sectional studies, one cannot exclude an inverted relationship or hidden confounding factors.
- Attempting to classify dietary style based on a single, 24-hour interview is insufficient to assess diet-semen. Seasonality, weekly deviations and compensatory behaviors are omitted.
- Interaction analysis assumes a strong statistical signal, but the groups with FSH < 8 IU/mL are small. Drawing conclusions about the strength of the diet relationship with such a limited n is illogical because the risk of a random effect is high.
- The text provides several post hoc tests between MEDAS categories and UPFs quartiles, but does not mention the FDR or Bonferroni control method. Interpreting single p<0.05 with such a number of comparisons can lead to false positives.
- A full hormonal profile (LH, testosterone, estradiol) is available only in 41–80 of 358 participants. Despite this, these are made about "preserved gonadal function" and an inferred interaction with FSH, which is inadequate for the range of data.
- Replacing the 14-point scale with three broad ranges (≤5, 6–9, ≥10) simplifies the analysis but may hide nuances (e.g. the difference between 5 and 3 points). This loses information about the potential dose-response.
Author Response
- Although the authors consistently repeat that these are only correlations, in several places in the discussion and abstract they formulate statements suggesting the effect of diet on semen parameters. However, in cross-sectional studies, one cannot exclude an inverted relationship or hidden confounding factors.
Answer
In agreement with the Reviewer’s concerns, we downsized conclusions emphasizing the associative nature of the findings.
- Attempting to classify dietary style based on a single, 24-hour interview is insufficient to assess diet-semen. Seasonality, weekly deviations and compensatory behaviors are omitted.
Answer
We agree with the Reviewer’s concerns. It should be noted that available studies addressing ultraprocessed food intake rely on a single assessment with a 24-Hour Food Recall Questionnaire. Accordingly, also considering the documented difficulty in maintaining adequate patient participation in the follow-up, the study protocol involved a single dietary assessment at basal. We added this elements in the discussion section.
- Interaction analysis assumes a strong statistical signal, but the groups with FSH < 8 IU/mL are small. Drawing conclusions about the strength of the diet relationship with such a limited n is illogical because the risk of a random effect is high.
Answer
We agree with the Reviewer’s concerns. Indeed, in order to adequately address the intraction with FSH in relation low number of patients in which this data is available, in the current version the logistic regression analysis has been specifically splitted into two models: the first excluding FSH and the second including FSH levels ≥ 8 IU/mL. Accordingly, we emphasized this limitation in the discussion section.
- The text provides several post hoc tests between MEDAS categories and UPFs quartiles, but does not mention the FDR or Bonferroni control method. Interpreting single p<0.05 with such a number of comparisons can lead to false positives.
Answer
We apologize for this missing. In the revised version of the manuscript we specificed that we adopted Bonferroni’s post hoc test for pairwise comparisons between subgroups in multivariate analysis.
- A full hormonal profile (LH, testosterone, estradiol) is available only in 41–80 of 358 participants. Despite this, these are made about "preserved gonadal function" and an inferred interaction with FSH, which is inadequate for the range of data.
Answer
As per the reply to point 3 of this Reviewer, we recognized the low number of patients with available hormonal data as a major limitation of the study. However, although limited, we cannot remove available hormonal parameters given their fundamental role in the physiology of spermatogenesis.
- Replacing the 14-point scale with three broad ranges (≤5, 6–9, ≥10) simplifies the analysis but may hide nuances (e.g. the difference between 5 and 3 points). This loses information about the potential dose-response.
Answer
In agreement with the Reviewer’s concerns, we now specified that subgrouping of MEDAS score into three broad ranges is not a matter of arbitraty classification but a recognized grading as previously repoprted [doi:10.3390/nu12102960]
Reviewer 2 Report (New Reviewer)
Comments and Suggestions for Authors
This study deals with na interesting topic in the field od human andrology as it has become widely accepted that a healthy and balanced lifestyle may have significant implications on the reproductive function in males.
Mediterranian diet has been acknowledged to have a significant positive effect on the overall fitness which is why the theme od the papier has merit and could be interesting for the readership of the journal.
The manusctipt reads well, I do not have major issues other than a few comments:
- was the intake of any nutritional supplements (vitamins, minerals or any supplementary formulas) taken into consideration during the evaluation?
- the participants could have been divided according to their age. Perhaps stronger or weaker age-dependent associations would be observed.
- The authors could briefly describe the MEDAS system for readers who may not be familiar with it. How does the evaluation work? How specific do the participants need to be?
- Isn’t a subjective assessment of habits a limiting factor affecting the outcomes?
Author Response
Was the intake of any nutritional supplements (vitamins, minerals or any supplementary formulas) taken into consideration during the evaluation?
Answer
We apologize for this missing information. In the revised version of the methods section, we specified that patients were at their first outpatient evaluation and, as per inclusion criteria, they were devoided of any current treatment with medications that could influence sexual and/or reproductive func-tion, current adherence to a nutritional intervention or completed within the previous 6 months, including dietary supplements,
The participants could have been divided according to their age. Perhaps stronger or weaker age-dependent associations would be observed.
Answer
We agree with the Reviewer’s suggestions. Indeed, all the described correlations between MEDAS score, or %Kal-UP, with semen parameters were maintained after correction for patient’s age (all p values < 0.001). We briefly added this result as data not shown.
The authors could briefly describe the MEDAS system for readers who may not be familiar with it. How does the evaluation work? How specific do the participants need to be?
Answer
In agreement with the Reviewer’s request, we briefly described that MEDAS questionnaire involves 14 items focused on the main food categories related to MD. MEDAS questionnaire was validated though the 136-item food frequency questionnaire (FFQ) and was recognized as a valid tool for the rapid estimation of MD adherence [doi: 10.3390/nu12102960].
Isn’t a subjective assessment of habits a limiting factor affecting the outcomes?
Answer
We absolutely agree with the Reviewer’s criticisms. Subjective assessment of habits is a major bias in any study involving questionnaires as assessment tools. However, the use of validated questionnaires widely recognized in available studies provides some limitations and may incur the above-mentioned risks.
Reviewer 3 Report (New Reviewer)
Comments and Suggestions for Authors
The manuscript reveals the possible association of nutrition with sperm parameters and male infertility. The authors should highlight the aim and the importance of their study, in other words what is the novelty of their findings.
They should also better describe the hormonal measurements and remake the tables 2 and 3, as they are difficult to understand, even though they refer to p-values.
In the discussion section a possible mechanism related to nutrition and male infertility, could be described, to better support the findings. They should also propose future directions about male fertility problems and nutrition.
The use of English language must be revised
Author Response
They should also better describe the hormonal measurements and remake the tables 2 and 3, as they are difficult to understand, even though they refer to p-values.
Answer
In agreement with the Reviewer’s suggestions, we corrected table 2 and 3 in order to provide a better comprehension of contents.
In the discussion section a possible mechanism related to nutrition and male infertility, could be described, to better support the findings. They should also propose future directions about male fertility problems and nutrition.
Answer
In agreement with the Reviewer’s suggestion, we briefly discussed the possible mechanistic basis linking mediterranean diet and spermatogenesis. We also discussed the use of well-designed clinical trials in order to address the possible role of nutrition as feasible approach to support male fertility.
The use of English language must be revised
Answer
In agreement with the Reviewer’s suggestion, we thoroughly revised English language.
Reviewer 4 Report (New Reviewer)
Comments and Suggestions for Authors
The manuscript “Role of Mediterranean Diet and Ultra-processed Foods on Sperm Parameters: Data From a Cross-sectional Study” provides information on the relationship between the type of diet and the improvement of sperm parameters. To be honest, I think that these basic results can contribute to the application of clinical research in the future. The data in the manuscript are interesting and prove what we thought about the relationship between diet and sperm quality.
- I understand that these foods affect sperm quality. However, to consume these foods regularly, we need a certain amount of income. What do the authors think is the relationship between wealth disparity and sperm quality (For example, how many people in Italy can sustain their intake of MD?)? I also think there are differences depending on race.
- In this study, many factors (hormones and sperm parameters) were examined to evaluate the relationship between diets and sperm quality. It is better to add a schematic model of which factors and how they are related to their functions.
- Do the authors think temporary intake of MD and UPFs will affect sperm quality?
The English could be improved to more clearly express the research.
Author Response
I understand that these foods affect sperm quality. However, to consume these foods regularly, we need a certain amount of income. What do the authors think is the relationship between wealth disparity and sperm quality (For example, how many people in Italy can sustain their intake of MD?)? I also think there are differences depending on race.
Answer
We are awared of importance of themes suggested by the Reviewer. However, this goes farely beyond the aims of he present study. Available data strongly support the role of ethnicity as a driving factor linking nutrition, endocrine function and fertility but, unfortunately, all patients enrolled were of caucasian ethnicity, although coming from different areas of Central and Eastern Europe. Finally, factors leading to the use of products attributable to the Mediterranean diet in Italy are not economic, since they are relatively inexpensive and easy to find, but rather cultural since they are associated with a family background. Differently, in other Countries, cheap food by definition is from fast food. However, once again, the discussion of these issues goes beyond the scope of the manuscript and, unless expressly requested by the reviewer, they will not be included in the revised text
In this study, many factors (hormones and sperm parameters) were examined to evaluate the relationship between diets and sperm quality. It is better to add a schematic model of which factors and how they are related to their functions.
Answer
In agreement with the Reviewer’s suggestion, we added a scheme, summarizing the relationship between diets and sperm quality, as a new Figure 6.
Do the authors think temporary intake of MD and UPFs will affect sperm quality?
Answer
It can be surely hypothesized an effect of temporary intake of MD or UPF on semen parameters. However, spermatogenesis takes about 3 months to be completed, one of which only on epididymis [doi: 10.1093/humrep/det020.]. Accordingly, in order to observe a possible consistent effect on semen parameters, the nutritional challenge, though temporary, should not least less than three months.
Round 2
Reviewer 1 Report (Previous Reviewer 2)
Comments and Suggestions for Authors
I have no comments
Reviewer 4 Report (New Reviewer)
Comments and Suggestions for Authors
I think that the revised manuscript has been fundamentally improved.
This manuscript is a resubmission of an earlier submission. The following is a list of the peer review reports and author responses from that submission.
Round 1
Reviewer 1 Report
Comments and Suggestions for Authors
This is an interesting study of sperm quality and adherence to a Mediterranean-style diet in a population of young men (average 35). The overall conclusion is that sperm quality is better if you consume a Mediterranean-type diet and few ultra-processed foods; which was expected. Less expected, however, is that this effect is independent of BMI.
My comments therefore aim to explore this contradictory conclusion that the effects are independent of BMI.
Table 1 should show the range of men entering the cohort (low values and high values plus median values) and the range of what is generally observed with such a population, particularly for hormonal values. Additional data of interest would be insulin and blood glucose levels.
As far as BMI is concerned, an average of 24 +/- 4 indicates that some men were of normal weight, but others were overweight and some could be obese. It's well known that obese men are hypogonadal, so I have my doubts about how these facts were taken into account in the statistical models.
As for the calories consumed, which are in the region of 2000+/- 300, meaning that these men don't eat too much, the question is why such a high BMI?
Table 2 should be presented differently to make it easier to read.
Author Response
Dear reviewer, thank you for giving us many thoughts to implement our work. We have discussed extensively all the points you rised in the discussion
We have updated tables 1 and 2 as as you suggested.
Relating to glucose and insulin we did not always have the data available because blood tests were not prescribed by our center. We decided not to consider the few that we had.
Regarding BMI, we have a very large sample and a low SD compared to an average still in normal weight, therefore it is plausible to think that those who were overweight or even obese (as well as underweight) were not representative. Having an average caloric intake of 2000 kcal (which is also what is considered an average reference consumption for an adult male) and a relatively small SD, supports the data of the sample on average in normal weight. Secondly, BMI is not directly attributable to caloric intake because the data itself is a ratio between weight and height which alone says nothing about energy expenditure if weight is not considered. Moreover, there is the intrinsic limitation of the 24-h recall method of food intake assessment that can lead to a very large error in estimating intake when compared with direct observation of the foods consumed.
Reviewer 2 Report
Comments and Suggestions for Authors
Comments to the Authors:
- line 138-The Authors state that "inclusion and exclusion criteria were defined for p < 0.05 and p > 0.1, respectively." Such wording is illogical, as inclusion/exclusion criteria should be based on population characteristics, not on statistical significance levels. The inclusion of p-values ​​in this context raises questions about the study methodology.
- lines 122–124 The Authors exclude participants who engage in physical activity for more than 3 hours per week. Such a procedure may introduce significant distortions - the patient group may not be representative of the general male population, and at the same time a factor that in itself may have a beneficial effect on semen quality is omitted. The lack of discussion on the impact of this decision on the generalization of the results is a substantive deficiency.
- lines 292–297 In the discussion section, the Authors suggest that better adherence to the Mediterranean diet correlates with reduced levels of FSH and LH, which would indicate a beneficial effect of the diet on the hypothalamic-pituitary-testicular axis. However, the mechanism of this relationship is not sufficiently explained, and the observations from the cross-sectional study do not allow for causality conclusions.
- lines 327–331 The lack of a deeper analysis of methodological limitations limits the substantive value of the conclusions.
- In the correlation and regression analyses, it can be noticed that some results (e.g. extremely high or illogical correlation coefficients) are not thoroughly discussed or interpreted. The lack of detailed commentary on why certain relationships, despite statistical significance, may be the result of measurement errors or other confounding variables, undermines the substantive value of the presented data.
Although the study addresses the interesting topic of the effect of diet on semen parameters, some elements of the methodology and interpretation of the results leave much to be desired. Ambiguous wording regarding the inclusion/exclusion criteria (line 138) and insufficient discussion of the mechanisms of the relationship between diet and hormonal and spermatological parameters result in a paper lacking substantive solidity.
Comments on the Quality of English Language/
Author Response
Dear reviewer thank you very much for helping us improve our work, we really appreciated your thoughts.
We realized that point 1. "line 138-The Authors state that "inclusion and exclusion criteria were defined for p < 0.05 and p > 0.1, respectively." Such wording is illogical, as inclusion/exclusion criteria should be based on population characteristics, not on statistical significance levels. The inclusion of p-values ​​in this context raises questions about the study methodology". Is an error related to an early version of the manuscript. We corrected the sentence.
We discussed all the other points you raised point by point in the work highlighting everything we changed as you suggested.
Thanks again for your time
Round 2
Reviewer 1 Report
Comments and Suggestions for Authors
no further comments, except that it would be useful to indicate the expected range for FSH, LH, SHBG, BMI, etc. levels in “normal” populations, in order to verify the representativeness of the cohort.
Comments on the Quality of English Languageno further comments, except that it would be useful to indicate the expected range for FSH, LH, SHBG, BMI, etc. levels in “normal” populations, in order to verify the representativeness of the cohort.
Author Response
no further comments, except that it would be useful to indicate the expected range for FSH, LH, SHBG, BMI, etc. levels in “normal” populations, in order to verify the representativeness of the cohort.
Answer
In agreement with the Reviewer’s request, we added reference values for semen and sex hormone parameters in table 1.
Reviewer 2 Report
Comments and Suggestions for Authors
In the revised version of the manuscript, there are still many important problems that make it not ready for publication. Even if the authors improved the language a little and added some new information, the article is still weak in the scientific idea, in the method, and in the way it is written.
First, the aim of the study is not clear. In lines 75–80, the authors say they want to check the relationship between the Mediterranean diet, UPFs, and semen quality — but they do not explain what kind of relationship they mean. Do they want to check correlation or causation? They need to write their hypothesis and questions more clearly.
There are also still many language mistakes. For example, in line 57 they write "kay" instead of "key". The sentence structures are also strange and not natural in English, like “have an incontrovertible essential role”. The text should be carefully checked again.
Methodological problems were not fixed. The authors used only one 24-hour recall to check what people eat. This is not enough to know their usual diet, especially with ultra-processed foods. They should use more recalls, or a food frequency questionnaire (FFQ). Another problem is that people with more than 3 hours of physical activity per week were excluded from the study, but the authors don’t explain why — and in fact, this kind of physical activity is often good for fertility.
In Table 1, some of the values have very big standard deviations — like sperm concentration 43.5 ± 42.3 — which shows that the group is very different inside. The authors do not say how they managed this variability, or maybe they should transform the data (e.g., use logarithms).
Another problem is that the authors talk about cause and effect — for example, saying the diet “improves” sperm — but this is a cross-sectional study. So they cannot say that diet causes something. They should say only “is associated with”.
There are also mistakes in the statistics part. In lines 133–137 they write “Person’s test” instead of “Pearson’s test”. This is a small thing but it shows that the text was not checked carefully.
One more problem is that we don’t know if the people who analysed the semen knew about the diet scores. If yes, that could be a source of bias. This should be explained.
Finally, the conclusions are too strong. In lines 33–36 and 243–251 the authors write that the diet may change spermatogenesis by changing hormones. But they don’t show real differences in hormones like TT, BAT or SHBG. These are only speculations and should be removed or written in a softer way.
Comments on the Quality of English LanguageI do not assess English.
Author Response
First, the aim of the study is not clear. In lines 75–80, the authors say they want to check the relationship between the Mediterranean diet, UPFs, and semen quality — but they do not explain what kind of relationship they mean. Do they want to check correlation or causation? They need to write their hypothesis and questions more clearly.
Answer
In agreement with the Reviewer’s concerns, we clarified that the primary aim of the study was to address the possible correlation between the adherence to the MD model, as opposed to UPFs in-take, and the quality of sperm parameters in subjects undergoing semen analysis.
There are also still many language mistakes. For example, in line 57 they write "kay" instead of "key". The sentence structures are also strange and not natural in English, like “have an incontrovertible essential role”. The text should be carefully checked again.
Answer
We apologize for these oversights. We now clarified that, in this scenario, the key role of nutrition in male fertility potential should be emphasized since the high degree of adherence to an healthy diet, such as the Mediterranean Diet (MD) model, has been associated better semen parameters and sperm function.
Methodological problems were not fixed. The authors used only one 24-hour recall to check what people eat. This is not enough to know their usual diet, especially with ultra-processed foods. They should use more recalls, or a food frequency questionnaire (FFQ). Another problem is that people with more than 3 hours of physical activity per week were excluded from the study, but the authors don’t explain why — and in fact, this kind of physical activity is often good for fertility.
Answer
We agree with the Reviewer’s concerns. To this regard, the single administration of the 24-Hour Food Recall Questionnaire is considered as representative for ultra-prodessed food intake. Repeated administration is typical of interventional studies, in which specific lifestyle/nutritional interventions are prescribed, to assess actual adherence to the prescription [DOI: 10.3390/nu17050835, DOI: 10.1017/jns.2025.6]. However, we strengthened these limitations in the study drawbacks.
In Table 1, some of the values have very big standard deviations — like sperm concentration 43.5 ± 42.3 — which shows that the group is very different inside. The authors do not say how they managed this variability, or maybe they should transform the data (e.g., use logarithms).
Answer
We agree with the Reviewer’s concern. We missed to clarify the use of Shapiro-Wilk test to address the normal distribution of data. The manuscript has now been corrected accordingly.
Another problem is that the authors talk about cause and effect — for example, saying the diet “improves” sperm — but this is a cross-sectional study. So they cannot say that diet causes something. They should say only “is associated with”.
Answer
We agree with the Reviewer’s concerns. Accordingly, we modified the discussion highlighting the associations
There are also mistakes in the statistics part. In lines 133–137 they write “Person’s test” instead of “Pearson’s test”. This is a small thing but it shows that the text was not checked carefully.
Answer
We corrected the text accordingly
One more problem is that we don’t know if the people who analyzed the semen knew about the diet scores. If yes, that could be a source of bias. This should be explained.
Answer
We specified that semen analysis was conducted blindly to the results of the dietary assessment.
Finally, the conclusions are too strong. In lines 33–36 and 243–251 the authors write that the diet may change spermatogenesis by changing hormones. But they don’t show real differences in hormones like TT, BAT or SHBG. These are only speculations and should be removed or written in a softer way.
Answer
We resized the study conclusions as requested.